# A Brief Review about the Role of Nanomaterials, Mineral-Organic Nanoparticles, and Extra-Bone Calcification in Promoting Carcinogenesis and Tumor Progression

**DOI:** 10.3390/biomedicines7030065

**Published:** 2019-08-28

**Authors:** Marina Senchukova

**Affiliations:** Department of Oncology, Orenburg State Medical University, 460000 Orenburg, Russia; orgma@esoo.ru or

**Keywords:** calcium carbonate, carcinogenesis, epithelial-mesenchymal transition, gastric cancer, nanoparticles

## Abstract

People come in contact with a huge number of nanoparticles (NPs) throughout their lives, which can be of both natural and anthropogenic origin and are capable of entering the body through swallowing, skin penetration, or inhalation. In connection with the expanding use of nanomaterials in various industrial processes, the question of whether there is a need to study the potentially adverse effects of NPs on human health becomes increasingly important. Despite the fact that the nature and the extent of damage caused depends on the chemical and the physical characteristics of individual NPs, there are also general mechanisms related to their toxicity. These mechanisms include the ability of NPs to translocate to various organs through endocytosis, as well as their ability to stimulate the production of reactive oxygen species (ROS), leading to oxidative stress, inflammation, genotoxicity, metabolic changes, and potentially carcinogenesis. In this review, we discuss the main characteristics of NPs and the effects they cause at both cellular and tissue levels. We also focus on possible mechanisms that underlie the relationship of NPs with carcinogenesis. We briefly summarize the main concepts related to the role of endogenous mineral organic NPs in the development of various human diseases and their participation in extra-bone calcification. Considering data from both our studies and those published in scientific literature, we propose the revision of some ideas concerning extra-bone calcification, since it may be one of the factors associated with the initiation of the mechanisms of immunological tolerance.

## 1. Introduction

Humans contact a large number of nanoparticles (NPs) throughout their lives. These particles are found in the atmosphere, the water, and the soil and can be of both anthropogenic and natural origin [1,2,3]. Interest in NPs is due both to their special properties and the rapidly developing nanotechnologies that use nanomaterials in various industrial processes, including for the production of the following: electronics, foods, pharmaceuticals, textiles, medical equipment, and others. For example, TiO_2_ NPs are used in food colorings, cosmetics, skin care products, and pigments for tattoos; Fe_2_O_3_ NPs are used for the final polishing of metal jewelry; ZnO NPs are added to many products, including cotton fabric and food packaging, thanks to their deodorizing and anti-bacterial properties [2,4,5,6,7]. 

The intensive development of nanotechnology increases the importance of the question of whether there is a need to study potentially adverse effects of NPs on human health, especially for those exposed under the high-risk conditions from professional activity and ecology. Currently, the assessment of the degree of risk associated with NP exposure is one of the most pressing questions asked of modern medicine. For this reason, the study of the characteristics of absorption, distribution, toxicokinetics, metabolism, points of application, mechanisms of toxic effects, and mechanisms and terms of the elimination of NPs from the body is attracted the attention of a huge number of scientists from around the world [1].

## 2. General Characteristics of Nano-Objects

It is important to note that nano-objects are an extremely heterogeneous group of substances. Figure 1 provides classification for main types of nano-objects, depending on their origin and their structure.

At present, it has been established that nano-objects are inherently more toxic than the substances of which they are composed and, further, they are more toxic than objects of micron size [1,8]. The toxicity of NPs is due to their physicochemical properties, the catalytic activity of their surface, and depends on the penetration route into the body, which can be inhaled, transdermal, transneural, and enteral. The sizes of NPs allow them to easily pass through the body’s biological barriers and accumulate into the internal organs, including the central nervous system [2,9]. The most important characteristics of NPs determining their toxicity are shape and structure, surface area, porosity, surface charge and catalytic activity, solubility in liquid media, and the ability to aggregate. These properties affect the absorption of NPs, their distribution in tissues and organs, toxicokinetics and metabolism, as well as the features of their biodegradation [2,9]. The most dangerous NPs in terms of their ability to produce pathological conditions are those in the size of up to 100 nm [10]. For example, particles having micron dimensions, if inhaled, settle in the upper respiratory tract, while inhaled NPs penetrate deeply and settle in the tracheobronchial and the alveolar regions, where they can cause severe respiratory disorders [11]. In the work of Huang (2017), it was noted that TiO2 NPs ranging in size from 10–30 nm were more toxic than larger ones due sharp increases in the formation of reactive oxygen species (ROS). In turn, positively charged NPs were more toxic than neutral or negatively charged ones, since both glucose-containing aminoglycans located on the cell membrane and DNA have a negative charge [2]. 

It is important to note that the small size of NPs allows them to penetrate into the underlying tissues, both through the cell membranes and between epithelial cells. When NPs penetrate the epithelial barrier, uptake by macrophages, lymphoid, and dendritic cells occurs, which spreads them to various organs and tissues. Once inside the cell cytoplasm, NPs can damage lysosomes and mitochondria, causing the formation of free radicals and ROS. NPs can also penetrate into the cell nucleus, causing DNA damage and gene mutations [1,9,12]. Some authors believe that the high biological activity of NPs is due to the fact that the size of NPs is identical to those of signaling molecules and cellular receptor sites as well as exosomes, all of which play important roles in intracellular communication and in the delivery of molecular signals from one cell to another [13]. In addition, according to several authors, the toxicity of NPs is characterized not only by physical and chemical properties but also by the biological or the protein “crown” that covers NPs after interacting with biological systems [14,15,16]. Protein crown masks NPs and largely determines their further fate. It is believed that “bio-corona” may be responsible for the recognition of NPs by immune cells for their biodistribution and elimination [1]. 

## 3. The Effect of Nanoparticles on the Respiratory System

The mechanisms governing the local impact of NPs to the respiratory organs have most extensively been examined using single-walled and multi-walled carbon nanotubes (CNTs), which are widely distributed in nature as a result of man-made, environmental pollution [1,12,17,18,19,20]. Studies on the respiratory toxicity of CNTs have shown that these nanomaterials cause a dose-dependent increase in biomarkers for cellular damage, initiate inflammation, cause damage to the lungs, and increase oxidative stress. After aspiration of CNTs, increases in the oxidation of protein sulfhydryls occurs, decreasing glutathione levels, depleting antioxidants, and increasing inflammatory markers and pro-inflammatory cytokines in bronchoalveolar lavage and in the lungs of experimental animals. Subsequently, multifocal granulomatous pneumonia and progressive interstitial fibrosis can develop [1,17]. Animals have also shown inflammatory changes in the heart and the liver as a result of an increase in lipid peroxidation products and the depletion of antioxidants. The addition of vitamin E to the diet of C57BL/6 mice could significantly reduce the severity of inflammatory changes and fibrosis in lung tissue.

## 4. Interaction of Ingested Nanoparticles with the Gastrointestinal Tract

Since NPs can be contained in food, medicine, and drinking water, they can enter the human body through the gastrointestinal tract (GIT). For example, NPs of SiO_2_, TiO_2_, Ag, and ZnO are added to food and other products, including toothpaste, cosmetics, and sunscreen [21,22]. In addition, inhaled NPs can penetrate the gastrointestinal tract, not due to their ingestion but through translocation from the systemic blood flow. For example, when rabbits were administered with Ag NPs intravenously, the particles accumulated not only in liver, kidneys, spleen, lungs, brain, testes, and thymus but also in the feces, which was an indication of their biliary excretion [23]. Li (2016) showed that, in rats that had inhaled CeO_2_ NPs, the total number of recovered NPs from extrapulmonary organs was much smaller than the amount recovered from the feces and the lungs [24]. The authors suggest that phagocytic cells can play a key role in the distribution of NPs that have entered the body through inhalation. These data indicate the relevance of studying the biological fate of NPs entering the body through the gastrointestinal tract and their effects on human health [25,26].

It should be noted that the anatomy of the gastrointestinal tract has a complex structure, and internal environment can significantly affect the properties of nanomaterials and consequently their biological activity. The properties of NPs can change under the influence of pH of gastric and intestinal secretions, their ionic and molecular composition, microflora, and the type and the quantity of food products with which NPs can interact [25]. Depending on the NPs size as well as physical and chemical properties, they can dissolve in the gastrointestinal tract (such as ZnO and Ag NPs [27,28]), they can undergo agglomeration (such as TiO_2_ [29,30]), and they can also release ions after contact with gastrointestinal fluids [25]. In the gastrointestinal tract, ingested NPs could undergo endocytosis by enterocytes, mucus producing goblet cells, and M-cells that deliver antigens to lymphoid structures, such as the Peyer’s patches and other gut-associated lymphoid tissue, thereby exerting a toxic effect on these cells. Translocation of NPs is believed to occur predominantly through the transcellular uptake route, while the paracellular route is not favored since epithelial cells are closely connected to each other through various contacts. In vitro and in vivo experiments showed that the internalization of NPs smaller than 100 nm occurs predominantly through clathrin-and/or caveoli-dependent endocytosis, while the larger particles underwent uptake by M cells via phagocytosis and micropinocytosis [30,31,32,33]. 

## 5. Toxic Effects of Nanoparticles on the Gastrointestinal Tract

To study the possible toxic effects of NPs on the gastrointestinal tract, both in vitro and in vivo studies were performed. 

### 5.1. In Vitro Studies of Nanoparticle Toxicity

It is suspected that toxicity of metallic nanomaterials is associated with the release of ions as a result of the dissociation of NPs under the influence of low gastric pH [25,28,34]. Overall, undifferentiated monocultures are more sensitive to the toxic effects of NPs than well-differentiated ones [35,36,37,38,39]. NPs of ZnO [36,39,40,41], SiO_2_ [38,42], and Ag [36,37] showed the highest toxicity in cell cultures. Moderate cytotoxic effects were noted for Au [43,44] and TiO_2_ [45,46] as well as CNTs [47]. For instance, the impact of TiO_2_ and Ag NPs on cell cultures was leading to loss of and morphological changes in microvilli, plasma membrane disruption, and genes expression changes, resulting in changes in tight junction anchoring [31,48,49,50]. 

Changes in cell cultures after exposure to NPs differed depending on chemical composition and concentration of the NP used, as well as the type of cell culture examined. Thus, a study of the cytotoxicity of TiO_2_ and CNTs on three human cell types showed that TiO_2_ NPs involved in the regulation of processes that were associated with inflammation, apoptosis, cell cycle arrest, DNA replication stress, and genomic instability. At the same time, the exposure of CNTs to cell cultures increased cell proliferation, DNA repair, and anti-apoptosis [47]. Another research studied the influence of various NPs in conditions of inflammation caused by IL-1β to the 3D intestinal model consisting of Caco-2 cells and two human immune cell lines. The authors noted that co-cultures released higher levels of IL-8 compared with Caco-2 monocultures [44].

It is worth mentioning, however, that not all researchers have observed toxic effects using TiO2 and other NPs under similar culture conditions [30,46]. It is believed that these conflicting findings may be explained by differences in doses, in vitro models, methods of detection, and physiochemical characteristics of the tested NPs [25].

### 5.2. In Vivo Study of Nanoparticle Toxicity

It should be noted that the available literature contains a limited number of publications concerning the study of the effects of NPs on the gastrointestinal tract in vivo. A number of studies on the toxic effects of NPs did not note when they were orally administered [51,52]. However, in other experiments, both local and systemic toxic effects of swallowed NPs were detected. Jeong (2010) showed that, in rats, 28-day oral exposure of Ag NPs, 60 nm in size, initiated a non-specific colitis, which was manifested by increased secretion of mucus in the ileum and the rectum, as well as changes in the composition of mucin [53]. NPs of TiO_2_ and CNTs, when taken orally, also caused inflammatory and even necrotic changes in the small intestine [54,55]. A number of studies have demonstrated the relationship between oral administration of TiO_2_ NPs and the development of colitis and colon cancer [50,56,57]. 

The systemic effects of NPs are due to the fact that the ingested NPs penetrate into the systemic circulation by endocytosis, where they cause inflammatory and oxidative damage to various organs, including liver, pancreas, kidneys, and spleen [58,59,60,61,62]. It is believed that smaller NPs can pass through the cell membranes of enterocytes, leading to a change in signaling or an increase in permeability and cytotoxicity, while larger NPs are absorbed predominantly by M-cells. It is known that M-cells play an important role in the development of the immunological tolerance to food and intestinal microflora, transporting genetic material, including proteins, inert particles, viruses, and bacteria of the intestinal lumen, to Peyer’s patches and other intestinal lymphoid tissues [63,64,65,66].

In the lumen of the digestive tract, NPs form complexes with bacterial peptidoglycan and lipopolysaccharides, but it remains unclear what effect, pro- or anti-inflammatory, is stimulated by these complexes [67,68,69]. When mononuclear phagocytes were co-cultivated with NPs conjugated with components of bacterial cells, increased expression of interleukin-1-beta and apoptosis-like cell death were observed, which indicates that NPs may have a proinflammatory effects [67]. On the other hand, Powell (2015) showed that endogenous calcium phosphate NPs, which are secreted into the lumen of the small intestine and interact with proteins, other food molecules, and microbiota to form mineral-organic NPs, are involved in the development of innate immunological tolerance to food and bacterial molecules [69]. 

### 5.3. Bioavailability of Nanoparticles

One of the main questions related to the possible effect of NPs on the gastrointestinal tract regards their bioavailability. In a rodent experiment, the bioavailability of NPs of TiO_2_ administered by gavage was shown to be 0.11% in the stomach and 4% in the colon, while the majority of the administered NPs accumulated in Peyer’s patches [70]. In a similar experiment, a bioavailability of Silica-Coated Upconversion Nanoparticles was also very low [71]. In a sense, these data can be extrapolated to people. In the volunteers, the ingestion of 100 mg TiO_2_ NPs led to the appearance of elemental Ti in the blood [72]. 

The presence of a biomolecular “crown” likely has significant impacts on the bioavailability of NPs. Formation of the “crown” is associated with the most important property of NPs—to adsorb biomolecules on their surface when in contact with foodstuffs and/or biological fluids. The physicochemical properties of biocorona can affect absorption, bioaccumulation, and biotransformation of NPs and can lead to unforeseen changes in the toxicity [14]. NPs can adsorb not only biomolecules but also polymers. Thus, Hinkley (2015) noted differences in the bioavailability of gold NPs depending upon whether they were uncoated or coated with polyethylene glycol (PEG). In the stomach, gold NPs without a PEG-coating formed large agglomerates of several hundred nanometers that did not change throughout the gastrointestinal tract. The PEG-coated gold NPs, however, retained their characteristics and were able to penetrate the mucin layer and were detected in tissues at a higher frequency than particles that lacked a PEG coating. However, in both cases, the bioavailability of the NPs was very low, reaching less than 1% [73]. Given that the absorption mechanisms of NPs in the gastrointestinal tract are not well understood in addition to the severity of this problem, open questions regarding the safety of various NPs (natural and man-made, for example, drugs) following oral exposure requires further research [1].

In summary, although the degree and the type of cell damage caused by NPs depends on the sizes as well as chemical and physical characteristics of the particles, a common mechanism related to toxicity is the ability of the particles to translocate to various organs through endocytosis and stimulate the production of reactive oxygen species. These lead to oxidative stress, inflammation, genotoxicity, and metabolic changes and could potentially lead to the formation of cancer [4,5,6,74,75,76].

## 6. Nanoparticles and Carcinogenesis

The fact that NPs can cause not only chronic inflammatory and autoimmune diseases but also malignant neoplasms has already been established [1,20,77]. The ability of asbestos nanofibers and CNTs to induce lung cancer and pleural mesothelioma following inhalation, for instance, has been demonstrated [77,78,79]. The authors noted that the effects of exposure to nanofibers depended less on the chemical composition of fibers than on size and geometry. The pro-inflammatory effects and the tendency toward the initiating fibrosis were most pronounced in cases where the size of the nanofibers did not allow macrophages to subject them to complete phagocytosis. Long and thin fibers were more toxic and carcinogenic than short, thick ones [78]. 

Besides the direct carcinogenic effects of NPs, some studies have shown that they can also influence tumor progression by stimulating metastases [18,80,81]. In rats, a single inhalation exposure to multi-walled CNTs induced the growth of metastatic lung tumors [18]. Further, chronic exposure to multi-walled CNTs caused DNA damage and increased mutation rates in human epithelial cells [82]. The exposure also induced apoptosis and the activation of major regulatory MAPK (mitogen-activated protein kinase) pathways, AP-1 (activator protein-1), NF-κB (nuclear factor kappa B), and Akt, all of which are associated with key molecular events involved in the formation of asbestos-induced lung cancer [83,84]. In addition, exposure to multi-walled CNTs resulted in the activation of myeloid-derived suppressor cells (MDSC), as well as increased serum levels of TGF-β1 and osteopontin (OPN) [18,20]. 

It has been established that contact of NPs with epithelial and immune cells results in the following carcinogenic effects:Increased synthesis of pro-inflammatory cytokines IL-2 (interleukin 2) and IL-6, IL-8, TNF-α (tumor necrotic factor-α), and NF-κB (nuclear factor kappa B) [17,20,38,50,54,57,85,86];Cell proliferative disorder as a result of activation PARP [Poly-(ADP-ribose) polymerases], AP-1 (Activator protein 1), NF-κB, Akt (Protein kinase B alpha), and MAPK (mitogen-activated protein kinase) [83,87];Release of large concentrations of reactive oxygen and nitric oxide having the free radical properties that damage cell membranes and genetic material [17,37,38,39,40,41,45,46,50,74,75,76,78,83];Development of immunological tolerance due to increases in TGF-β1serum levels [17,19,20,54];Generation of CD4+CD25+FoxP3+ regulatory T-cells [88,89,90] and PD-L1 (ligand 1 of programmed cell death protein) activation [91,92].

It is important to note that similar results were obtained when studying NPs with different chemical compositions. These data indicate that the observed pro-carcinogenic effects of NPs can be associated not only with their structure and, above all, their size. The main pro-carcinogenic effects of nano-objects are presented in Figure 2.

In addition to the above effects, the effect of “extravasation induced by NPs” has recently been described by scientists. In vivo studies have shown that the intravenous administration of some NPs led to the disorder of endothelial cells adhesive properties and increased the vessel permeability. In particular, this effect was observed in experiments with the intravenous administration of Au, Si, TiO_2_, and other NPs sized 10–40 nm. It is believed that the effect may be useful in the treatment of malignant tumors, as it may facilitate the release of pharmaceuticals from the bloodstream and allow for their direct delivery to tumor cells [93,94]. However, in a recent experimental model of breast cancer, it was shown that intravenous injections of TiO_2_, SiO_2_, and Ag NPs significantly increased intravasation and extravasation of tumor cells, thereby contributing to the active appearance of new metastatic foci [95].

One of the principal pro-carcinogenic effects arising from the contact of NPs with various cell types is an increase in TGF-β expression. This effect has been noted by many researchers [18,19,20,85]. It is known that TGF-β is a multifunctional cytokine, the main functions of which are associated with the regulation of proliferation, differentiation, motility, and adhesion of various cells, as well as with participation in the processes of angiogenesis, immunological tolerance, and cancer metastasis. A high level of TGF-β expression is observed in various inflammatory, autoimmune, and oncological diseases [96,97]. Cytokine activates pro-invasive and prometastatic immune responses through Smad, Snail, NF-κB, Wnt, and Ras signaling pathways [98,99,100]. In response to certain immune stimuli, TGF-β inhibits the differentiation of cytotoxic T-lymphocytes, Th1 and Th2 cells, and stimulates the formation of peripheral T-regs (regulatory T-lymphocytes), Th17, Th9, and Tfh cells [97]. The generation of T-regs is characterized by the expression of CD25 and the Foxp3 transcription factor [101]. T-reg cells express TGF-β, contributing not only to the suppression of an excessive immune response [102] but also to the activation of the mechanisms of epithelial-mesenchymal transition (EMT) [103]. Several studies have shown that increased expression of TGF-β and markers of EMT are associated with the activation of PD-L1 expression in the tumor microenvironment, mainly on T-regs lymphocytes, macrophages, and dendritic cells [104,105,106]. Also, the association of TGF-β expression by tumor cells with the development of resistance to various anticancer drugs has been noted [107]. Currently, TGF-β is considered as one of the key markers associated with immunological tolerance and as a target for antitumor immunotherapy. For example, the use of the TGF-β2-targeting antisense molecule trabedersen (AP12009) contributed to improving the survival rate of patients with skin melanoma [108].

## 7. Endogenous Nanoparticles and Their Role in Physiological Processes and Pathology

When studying the literature concerning endogenous mineral organic NPs, we noted that researchers often use different terms when describing structures similar in their characteristics. For example, particles having similar sizes, chemical composition, and properties were called “mineral organic nanoparticles” [26,109,110], “calcifying nanoparticles” [111], and “calciprotein particles” [112,113]. Because of this, it is not always clear in the literature whether the particles described refer to different or the same nanostructures. This is an important factor to consider when studying the role of NPs in health and pathology.

It should be noted that endogenous mineral-organic NPs are found in practically all human body fluids. The authors describe them as spherical and ovoid particles with a diameter of 50–500 nm [109,110,114]. It is believed that their formation may be associated with exosomes, extracellular membrane vesicles with a diameter of 30–100 nanometers, secreted into the extracellular space by cells of various tissues and organs [115]. The exosome cavity is of cytoplasmic origin and contains proteins, lipids, DNA, and various types of RNA, including mRNA, microRNA, and long non-coding RNA. The membrane of exosomes is formed as a result of invagination inside the endosomal membrane. It has been established that exosomes are involved in intercellular communication—the transfer of genetic material from one cell to another—and facilitate the immune response through presentation of antigens. They are found in various biological fluids of the body, such as serum, cerebrospinal fluid, urine, saliva, and breast milk [116]. The level of exosomes and mineral-organic NPs is elevated in the body fluid of people suffering from various diseases; therefore, some authors have suggested that they may be involved in the development of various pathological conditions, including arthritis, atherosclerosis, cancer, and chronic kidney disease. Studies have shown that, in malignant neoplasms, exosomes can participate in the reprogramming of cancer cells from an epithelial to a mesenchymal phenotype, thereby promoting invasion, metastasis, and drug resistance of tumors [117,118,119,120]. 

Initially, endogenous mineral NPs were regarded as nanobacteria, and there was a belief that they were the smallest form of living microorganisms and were associated with the occurrence of various human diseases [121,122]. Later, it was shown that the described NPs are non-living mineral particles that mimic living microorganisms in various ways; for example, they have similar morphologies, the ability to increase in size and particle number in culture, and the ability to bind with biological molecules (carbohydrates, lipids, metabolites, nucleotides, and proteins) [109,110,123,124]. Similar processes occur in inanimate nature. Wu (2016) demonstrated that mineral particles from 20–800 nm in diameter that are formed in sea, spring, and soil water have a round, oval, or irregular shape [26]. They are characterized by a pronounced tendency to aggregate, resulting in the formation of structures resembling coccoid bacteria. Some of the structures formed are very similar to bacteria undergoing cell division. Individual samples can form film-like structures. The authors found that the formation of round NPs in studied samples was possible since mineral particles found in surface waters bind to organic molecules, forming mineral–organic complexes.

In regard to the chemical composition of mineral-organic NPs found in various biological human fluids, CaCO_3_ and Ca_3_(PO4)_2_ are most abundant [109]. When mineral NPs are cultivated in biological fluids, they contact with proteins, which give them a rounded shape [125]. This form makes them appear similar to previously described nanobacteria [121,122].

It is important to note that CaCO_3_ is a very abundant mineral in nature. Drinking water contains a large amount of polydisperse NPs and solid materials of irregular shape, consisting mainly of CaCO_3_ and CaSO_4_, often with the addition of other elements such as iron oxides [3,26]. CaCO_3_ nanoparticles can be formed by chemical and physical processes such as weathering, dissolving, and precipitating carbonates under the influence of CO_2_ with the intermediate formation of bicarbonate [3]. Chin (1998) showed that, in seawater, CaCO_3_ interacts with organic substances of plant and animal origin and participates in the formation of a colloidal gel, where the CaCO_3_ is found in the lattice sites [126]. The concentration of the mineral inside the gel is higher than in the surrounding water and, under certain conditions, it can crystallize and precipitate. It is believed that similar processes can occur in body tissues, leading to the formation of microcalcinates. The main trigger of this process is a change in pH.

The importance of considering the possible effect of CaCO_3_ nanoparticles on biological processes in health and disease is also explained by the fact that CaCO_3_ nanomaterials are widely used in food and pharmaceutical industries [127,128,129,130]. A number of authors have noted its very low toxicity during oral administration [131] and when cultured with cell cultures [132]. However, other researchers have pointed out its potential cytotoxicity, which applies not only to normal cells but also to tumor cells. Thus, Zhang et al. (2014) noted a pronounced cytotoxic effect of CaCO_3_ NPs to breast cancer cell line MDA-MB-231, which was manifested by a change in the size and the morphology of cells, the formation of large cytoplasmic vacuoles, the inhibition of proliferation, and the induction of apoptosis [133]. After internalization of CaCO_3_ NPs by cancer cells, decrease in cell size, chromatin condensation, fragmentation, and dissolution of the nucleus with the formation of apoptotic bodies were observed. These features of CaCO_3_ NPs served as the basis for the development of new pharmaceuticals for the treatment of cancer [134,135,136]. It is assumed that CaCO_3_ NPs can be used for the delivery of chemotherapy drugs that will enhance their antitumor activity. For example, the use of CaCO_3_ in a breast cancer cell culture together with doxorubitsin showed a more pronounced antitumor effect than the use of doxorubicin alone [137].

Another important property of CaCO_3_ NPs is their ability to penetrate not only the cytoplasm but also the nucleus of cells. Zhao (2014) and Wang (2014) showed that CaCO_3_-based NPs can be very effective for delivering plasmid DNA to the genome. Addition of calcium phosphate or protamine sulfate to the structure of NPs significantly increased the efficiency of absorption and transfection of plasmid DNA [138,139].

The cytotoxicity of CaCO_3_ NPs has also been noted in vivo. With subcutaneous administration of CaCO_3_ NPs, a maximum cytotoxicity was observed in the acute toxicity group (single subcutaneous administration at a dose of 29,500 mg/m^2^) and in the high dose group (daily subcutaneous administration at a dose of 5900 mg/m^2^ for 28 days). In animals of these groups, granular lesions in the liver and congestion of the heart and the kidneys were observed. The kidneys showed multifocal interstitial polymorphonuclear infiltration. There were vacuolar degenerations and necrosis of renal tubules. Animals also exhibited generalized congestion and had exudates in the lungs [140]. 

A number of studies also demonstrated the ability of CaCO_3_ to indirectly influence mechanisms of tumor progression. When CaCO_3_ NPs were co-cultivated with various cell cultures, cytotoxicity of the particles was manifested by endocytosis, the production of intracellular ROS, membrane damage, and cell apoptosis [141,142]. Similar results were obtained by Peng [110] and Horie [143]. The joint cultivation of CaCO_3_ NPs with fibroblasts, despite their low cytotoxicity, led to an increase in TGF-β1, VEGF levels, and cell proliferative activity [144].

Of interest is the work of Powell (2015), who described the mechanism for the formation of immunological tolerance to food and the intestinal microbiota with the participation of endogenous, calcium-containing, mineral-organic NPs [69]. The authors showed the possibility of endogenous formation of calcium phosphate NPs from calcium and phosphate ions, which naturally secreted in the distal small intestine. In the lumen of intestine, NPs trap soluble protein macromolecules of endogenous and exogenous origin and transport them to the Peyer’s patches via M-cells, thereby initiating immune cells to PD-L1 expression (immune tolerance-associated molecule). We previously showed that the induction of gastric cancer in rats could be accomplished by using a mixture of formaldehyde and hydrogen peroxide [145]. Adding a suspension containing CaCO_3_ NPs to this mixture resulted in sizable increases in its carcinogenic properties, which were manifested by a reduction in the number of carcinogen administrations and the time until gastric cancer induction. Microscopically, the tumors were intramucosal carcinoma with extensive invasion of tumor cells into the muscle layer, the serosa, and the omentum. The tumor cells were positive for vimentin, Snail, and TGF-β2, which testified to the activation of the mechanisms of epithelial-mesenchymal transformation. Multiple microcalcifications were detected in gastric mucosa and mesenteric lymph nodes of the experimental animals. In control rats, the described changes were not observed [146]. Thus, considering the above data in combination with the scientific literature, it is likely that, under the conditions of inflammation or carcinogenesis, CaCO_3_ NPs may affect the activation of mechanisms of EMT and immunological tolerance. We also assume that endogenous calcium containing mineral-organic NPs, by adsorbing antigens, can play a key role in the delivery of various antigens to immune organs, participating in the formation of immunological tolerance, including to tumor cells.

To conclude this section, we give a table that summarizes the main pro-carcinogenic effects of various NPs (Table 1).

## 8. Mineral-Organic Nanoparticles and Extra-Bone Calcification

As noted above, calcium containing mineral-organic NPs can play a key role in ectopic calcification [79]. It should be noted that extra-bone depositions of calcium salts are widespread in the human body; however, the mechanisms of this process and its role in the development of pathological conditions remain unclear. There are two types of extra-bone calcification, a metastatic form associated with hypercalcemia and a metabolic form caused by metabolic disorders in tissues. Metabolic calcification is observed in atrophic, dystrophic, dysplastic, and necrobiotic changes in tissues, as well as in benign and malignant tumors [148,149,150]. In malignant neoplasms, micro and macrocalcinates have been detected in cancers of the breast, the kidney, and the thyroid, as well as in some other tumors [148,149,150,151,152,153].

It is known that biological fluids contain factors that slow down or inhibit the formation of CaCO3 crystals, contributing to the formation of round amorphous structures that were previously incorrectly interpreted as nanobacteria. Serum contains powerful inhibitors of the spontaneous precipitation of calcium and apatite [154], and these include both calcium-binding proteins such as albumin and apatite-binding proteins such as fetuin-A [155]. It is believed that the binding of calcium salts by serum proteins is aimed at inhibiting excessive calcification [156]. However, when the maximum concentration of mineral NPs is exceeded, they can precipitate to form microcalcifications [79]. Wong (2015) showed that mineral NPs containing the serum proteins albumin and fetuin-A were initially deposited in the form of round amorphous NPs, which gradually increased in size, aggregated, and combined, forming crystalline mineral films similar to the structures observed in calcified human arteries [114]. It is believed that exosomes can initiate ectopic calcification in the human body [116]. At the same time, some experiments have shown the innate ability of tumor cells to form microcalcifications [147]. The biological significance of this process remains unclear.

One of the principal pathogenetic processes associated with metabolic calcification is chronic inflammation [157,158]. However, the question of what comes first, inflammation or pathological calcification, remains open. Peng (2013) showed that calcium-containing serum NPs could be phagocytosed by macrophages, inducing the production of mitochondrial ROS, the activation of caspase-1, and the secretion of interleukin-1β [110], thereby promoting inflammation. At the same time, Kumon (2014) explored the calcium-containing NPs derived from urinary stones (P-17) using anti-P-17 IgM monoclonal antibodies specific for oxidized lipids and was able to determine that these NPs were a by-product, not an etiological agent, of chronic inflammation. The authors showed that the lamellar structures of NPs consist of acidic/oxidized lipids that provide structural frameworks for carbonate apatite. They believe that lipid peroxidation can be the main cause of the production of calcium-containing NPs, and oxidized lipids can be a common platform for ectopic calcification in atherosclerosis-prone (ApoE−/−) mice [123].

It has now been established that the extra-bone deposition of calcium salts is associated with increased expression of pro-inflammatory cytokines, alkaline phosphatase, and bone-related proteins such as osteopontin and osteoprotegerin, with an increase in TGF-β levels, as well as with the trans-differentiation of smooth muscle cells into osteoblastic-like cells [158,159]. Interestingly, the increase of bone-related protein levels, on the one hand, potentiates the processes of extra-bone calcification [159,160]; on the other hand, it activates the mechanisms of EMT in various pathological and physiological processes [161,162]. A number of studies have noted a direct link between the extra-bone deposition of calcium salts and the activation of EMT mechanisms [151,163]. These data suggest that the formation of microcalcifications in tumor tissue may be associated with the activation of EMT and, therefore, with tumor progression. Indeed, the presence of microcalcifications in tumors has been shown to be associated with an adverse prognosis in breast [151,163], thyroid [157], and kidney cancers [164]. Moreover, the subcutaneous administration of calcium oxalate into the dairy inguinal crease induced the development of breast cancer in mice [147].

In conclusion, we would like to draw attention to another fundamental question concerning the assessment of the role of extra-bone calcification in various pathological processes. What are the common particulars of atrophic, dystrophic, dysplastic, and necrobiotic changes in tissues, as well as benign and malignant tumors? What connects these seemingly completely different processes? We think that the appearance of cells with genetic damage, which essentially represent autoimmune heterologous material, could unite all these processes. If there were no mechanisms for blocking autoimmune responses, their development would inevitably lead to death of the organism. In this sense, the development of immunological tolerance to its own altered cells is the most important mechanism for preserving the integrity of a living organism. According to modern concepts, the main role in the development of acquired immunological tolerance is given to the activation of T-regs, followed by the synthesis of inhibitory cytokines: TGF-β, IL-10, IL-35, and others [97]. However, at present, it is not clear what serves as a release mechanism in the development of immunological tolerance. It might be anticipated that the activation of the processes of extra-bone calcification may be one of the factors associated with the initiation of the mechanisms of immunological tolerance. The fact that the same cytokines, mediators, and transcription factors are involved in the processes associated with both extra-bone calcification and the development of immunological tolerance indirectly supports this hypothesis [157,158]. We believe that further research is needed to confirm it.

## Figures and Tables

**Figure 1 biomedicines-07-00065-f001:**
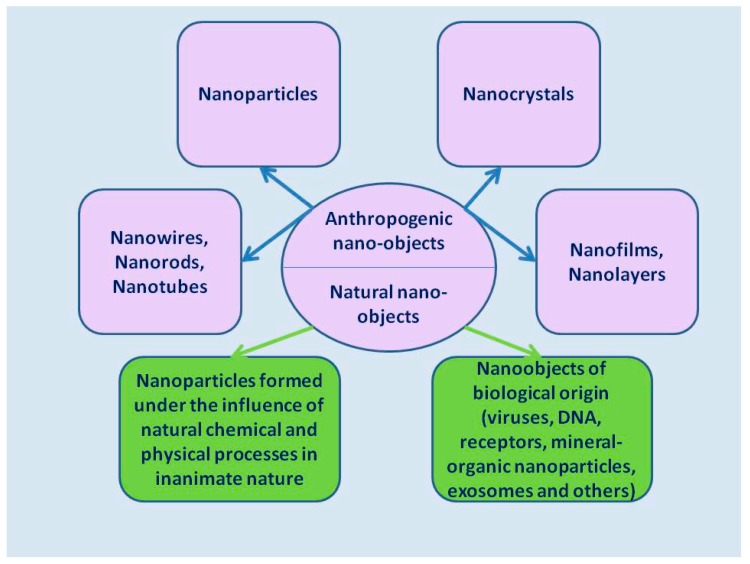
The main types of anthropogenic and natural nano-objects.

**Figure 2 biomedicines-07-00065-f002:**
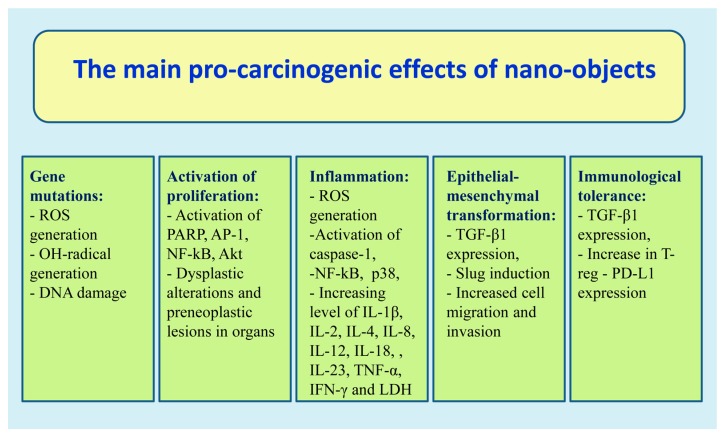
The main pro-carcinogenic effects of nano-objects.

**Table 1 biomedicines-07-00065-t001:** The main pro-carcinogenic effects of different types of nano-objects.

Nano-Objects	Size (nm)	Cell Type/Animal/ Features of Experiment	Main Effects	Reference
**In Vitro Studies**				
SWCNT	228 ± 77	Pulmonary MDSCs from SWCNT-exposed Wild type mice bearing LLC – co-cultured with T-cells Pulmonary MDSCs – in LLC-conditioned medium	Increase in IL-2 production; Suppression of syngeneic T cells proliferation Increase in TGF-β1 production	[19]
SWCNT	D1–4 x L100–1000	Murine macrophage cells (RAW 264.7) and murine lung epithelial cells (MLE-15)	Increasing level of LDH, OPN, and TGF-β; Inhibition of OPN production by anti-OPN antibody reduced the level of TGF-β1	[20]
SWCNT	D0.8–0.2 x L100–1000	Human bronchial epithelial cells (BEAS-2B)	Induction of malignant transformation in the cells with the initiation of cancer stem-like cells (CSCs); Activation of EMT via the Slug induction; SWCNT-transformed cells exhibited aggressive cancer phenotypes, including increased cell migration and invasion; Subcutaneous injections of SWCNT-transformed cells into nude mice led to the formation of malignant tumors capable of metastasis	[81]
SWCNTs	D0.8–2.0	Normal mesothelial cell (NM) and malignant mesothelial cell (MM) cultures	OH-radical generation and production of ROS; Induction of DNA damage; Activation of PARP, AP-1, NF-kB, p38, and Akt	[83]
MWCNT vs. TiO_2_ nanobelts	458 ± 16 634 ± 86	Human macrophages (THP-1), SAE and intestinal (Caco-2/HT29-MTX) cells	A low level of toxicity for MWCNT; The common response of all three types of cell cultures to TiO_2_ NPs exposure was the activation of genes transcription responsible for apoptosis, inflammation, antigen presentation, angiogenesis, and epithelial-to mesenchymal transition	[47]
Graphene quantum dots	N/A	Monocyte-derived DCs, Human peripheral blood mononuclear cells – magnetic-activated cell sorting Mixed cell cultures – co-cultivation DCs and T cells	Decreased T-cell proliferation and Th1 и Th17 differentiation; Induction of suppressive CD4+CD25+FoxP3+ regulatory T-cells; Decreased production of ROS	[89]
CNTs, GNFs	D1.1 x L500–100,000 D30–50 x L500–20,000	Human bronchial epithelial BEAS 2B cells	Initiation of DNA damage and increased mutation rates in cells	[82]
CuO	10	Undifferentiated and differentiated Caco-2 intestinal cells	Concentration dependent decrease in cell viability in undifferentiated cells; IL-8 production was over 2-fold higher in undifferentiated cells; Tight junction dysfunction and decreased barrier integrity	[35]
ZnO	N/A	Chinese hamster lung fibroblast cells (V-79)	Decrease in cell viability and an increase in ROS; Increase in frequency of HGPRT gene mutation	[76]
Aminated polystyrene amine, ZnO, Ag	17 ± 2, 107 ± 45	HeLa cells	Induction of cell apoptosis and release of ROS	[36]
ZnO vs. Ag	90	Caco-2 cells	ZnO NPs exerted higher cytotoxicity than Ag NPs; Significant depletion of superoxide dismutase level and release of ROS	[37]
ZnO vs. TiO_2_	N/A	Caco-2 cells	Generation of ROS; Increase in IL-8 secretion; DNA damage by ZnO; ZnO NPs was more toxic than TiO_2_ NPS	[40,41]
TiO_2_	30–50	Human intestinal epithelial cells (IECs) and macrophages	Release of pro-inflammatory cytokines IL-1β, IL-18; The production of ROS and increasing epithelial permeability in IEC monolayers	[50]
SiO_2_	15, 55	Caco-2 cell	Cell death and chromosome damage; Generation of ROS; Increase in IL-8 secretion; SiO_2_-15 nm was more toxic than SiO_2_-55 nm	[38]
CaCO_3_	40–60	Mouse embryonic fibroblast NIH 3T3 cell line	At concentrations of 200 to 400 µg/ml, a slight decrease in cell viability and increase in ROS generation and LDH levels	[132]
Biomimetic calcium phosphate NPs	<100	Neutrophils and macrophages isolated from whole blood of volunteers, Co-culture of neutrophils and macrophages	Production of mitochondrial ROS; Activation of caspase-1; Secretion of IL-1β	[110]
Food nano CaCO_3_ NPs vs. Food bulk CaCO_3_ NPs vs. reagent CaCO_3_ NPs (SS CaCO_3_)	100 2000 110	Human intestinal epithelial (INT-407) cells	Food nano CaCO_3_ exhibited the highest cytotoxicity in terms of ROS generation, membrane damage and LDH release	[142]
CaCO_3_	N/A	Human lung carcinoma A549 cells, Human keratinocyte HaCaT cells	Small increasing ROS level; Increase in C/EBP-homologous protein (CHOP) expression and the activation of caspase-3	[143]
CaCO_3_	35–60	MC3T3-E1 and hFOB 1.19 osteoblast cell lines	CaCO_3_ NPs exhibit a low cytotoxicity and genotoxicity; Increase in TGF-β1, VEGF levels and cell proliferative activity	[144]
CaCO_3_/CaP/ DNA vs. CaCO_3_/ DNA	N/A	293T cells, HeLa cells – pGL3-Luc plasmid was used for gene transfection	Cellular uptake and nuclear localization of CaCO_3_/CaP/DNA NPs were significantly enhanced as compared with CaCO_3_/DNA NPs	[138]
Protamine sulfate -calcium carbonate-plasmid DNA (PS-CaCO_3_-DNA) NPs	N/A	293T cells, HeLa cells – pGL3-Luc and pEGFP-C1 plasmids were used for gene transfection	PS-CaCO_3_-DNA nanoparticles were more effective in gene delivery than both PS-DNA NPs CaCO_3_-DNA NPs	[139]
**In Vivo Studies**				
MWCNT	D10 x L18 000	C57BL/6J mice – MWCNT aerosol	Bronchioloalveolar inflammation; Hyperplasia, hypertrophy and metaplasia of the bronchiolar epithelium, lung fibrosis; Vascular changes by type of vasculitis	[12]
MWCNT	N/A	C57BL/6 mice – inhalation exposures	Pleural granulomas formation; The release of cytokines and oxidants which damage of the mesothelial and endothelial cells, enhance inflammation, fibrosis and genotoxicity; Pro-inflammatory and pro-carcinogens effects of MWCNT were observed only when the nanotubes were long and thin	[78]
SWCNT	D100 x L1000	C57BL/6 mice – inhalation exposures vs. pharyngeal aspiration exposures	Increasing level of LDH, TGF-β1, TNF-α and IL-6 in BAL; Increase in concentration of lipid peroxidation products in lung homogenates; Significant depletion of total antioxidant status in lung homogenates; *K-ras* mutation in lung; SWCNT inhalation is more toxic than aspiration	[17]
CNF vs. SWCNT vs. Asbestos	D80–60 x L5000–30,000 D65 x L1000–3000 D160–800 x L2000–30,000	C57BL/6 mice – pharyngeal aspiration	Induction of chronic bronchopneumonia, pulmonary fibrosis and lymphadenitis; Genotoxic effects and increase in the incidence of *K-ras* oncogene mutations in the lung; Inflammation was more severe in asbestos- and CNF-treated mice whereas the severity of fibrotic and genotoxic effects - in SWCNT-treated mice	[86]
Porous silicon NPs	200	C57BL/6 mice, Common marmosets (Callithrix jacchus) – intravenous injection	Increase by 5-fold in the number of splenic CD4+CD25+FoxP3+ regulatory T-cells compared to control mice	[90]
PLG(Ag)	450–850	Mouse model (SJL/J mice) of EAE – subcutaneous injection of PLP and after 7 days intravenous injection of PLG+PLP	Complete prevention of EAE after intravenous administration of PLG+PLP; Significant increase in PD-L1 expression in Kupffer cells, macrophages and dendritic cells of hepar	[92]
Ag, Au, Fe_3_O_4_, SiO_2_, ZnO, CuO, NiO, MnO, PbO, Al_2_O_3_, TiO_2_	3.4–1000	Outbred white rats – intratracheal instillation – intra-peritoneal injections of the same during 6–7 weeks	Ultrastructural abnormalities in cells of the liver, spleen, kidney, myocardium, brain, thymus, and testicle tissues did not depend on the NPs type; Cytotoxicity manifested by vacuolization of the cytoplasm, damage of mitochondria with partial or complete loss of cristae; Genotoxic effect	[8]
TiO_2_	30–50	C57BL/6J and NLRP3-deficient mice – model of dextran sodium sulfate-induced colitis (DSS-treated mice) – by oral gavage administration	A more severe colitis with a significant shortening of the colon; A higher inflammatory cell infiltration with a severe disruption of the mucosal epithelium in TiO2-treated mice	[50]
TiO_2_	66, 260	Bl 57/6 male mice – by oral gavage administration	Increase in the levels of IL-12, IL-4, IL-23, TNF-α, IFN-γ, and TGF-β in samples of jejunum and ileum; Increase in the levels of T CD4+ cells in duodenum, jejunum, and ileum	[54]
TiO_2_	300	BALB/c male – colitis associated cancer (CAC model - DSS-treated mice) – by oral gavage administration	Dysplastic alterations in the distal colon; Increase in the levels of tumor progression markers in the small intestine	[56]
TiO_2_ (E-171)	80–100	Wistar rats – by oral gavage administration or with drinking water – induction of colon carcinogenesis by 1,2-dimethylhydrazine	Increase in the number of preneoplastic lesions in colon; Significant increases in TNF-α, IL-8, and IL-10 levels in the colonic mucosa of E171-treated rats without activation of caspase-1	[57]
TiO_2_	33, 160	CBAB6F1 mice – by oral administration	Induction of DNA-damage in the cells of bone marrow and liver; Increase in the mitotic index in forestomach and colon epithelia, and apoptosis in forestomach and testis	[58]
TiO_2_	14–50	Balb/c mice to – transdermal exposure	Increase in ROS generation, levels of immunoglobulin E, IL-8, 8-hydroxy-2′-deoxyguanosine, soluble intercellular adhesion molecule-1, and C-reactive protein.	[75]
Ag	60	Sprague-Dawley rats – by oral administration	Initiation of a non-specific colitis which increased secretion of mucus in the ileum and rectum	[53]
Ag-polymer conjugate NPs	80, 400	SJL/J mice, C57BL/6J mice – a subcutaneous administration	Increase in generation of CD4+CD25+FoxP3+ regulatory T-cells by BMDCs that were generated from the bone marrow of C57BL/6J mice treated with NPs	[88]
CaCO_3_	30 ± 5	Sprague-Dawley rats – a single subcutaneous administration at a dose of 29,500 mg/m^2^ – a daily subcutaneous administration at a dose of 5900 mg/m^2^ for 28 days	Anorexic, dyspnoeic, fever, tachycardia and a serious gangrene lesion; Increased in levels of ALT, ALP, AST, bilirubin, urea, and creatinine; Granular lesions in the liver, congestion of the heart and the kidneys; Multifocal interstitial polymorphonuclear infiltration and vacuolar degenerations and necrosis of renal tubules in kidneys; Generalized congestion and had exudates in the lungs	[140]
**Cancer Induction**				
MWCNT	D30–80 x L500–5000	B6C3F1 mice – intraperitoneal injection of MCA for carcinogenesis promotion and one week after that - the MWCNT inhalations	Lung cancer or bronchiolo-alveolar adenoma were in 11% of mice - in air group, in 18% - in MWCNT, in 33% - in MCA and in 76% - in MCA + MWCNT group; Five mice (9%) exposed to MCA and MWCNT and 1 (1.6%) exposed to MCA also developed tumors morphologically corresponding to sarcomatous mesotheliomas	[80]
MWCNT-7	D30–80 x L2500, D30–80 x L4200, D100 x L5000	F344 rats and B6C3F1 mice – intratracheal instillation in rats – aerosol inhalation in mice + MCA – a single intraperitoneal injection in rats and mice –subcutaneous injection	Results are presented for maximum NPs concentrations Pleural mesothelioma - in 15.8% rats, lung carcinoma – in 36.8%, 0% - in control group Lung carcinoma in 22% mice in MCA group, in 62% in MCA+MWNT-7 Peritoneal mesothelioma in 87.5% in mice, in 87,5% in rats, 0% in control group No tumor induction	[77]
MWCNTs	D40–90 x L4000	F344 rats -aerosol inhalation	22% and 16% cases of lung cancer in males and females. Lung carcinomas were mainly bronchiolo-alveolar carcinomas	[79]
SWCNT	230	C57BL/6 mice and TGF-β-deficient mice –model of LLC –pharyngeal aspiration	Increase in the number and size of tumor nodules in the lung; Increase in numbers of MDSC in lymphoid tissues, lung, spleen and bone	[18]
Calcium oxalate	N/A	BALB/c or BALB/c nude mice –7 injection in the mammary fat pad area in a period of 18 days	All mice had breast cancer on day 20 of experiment	[147]
CaCO_3_	7.8 ± 10.8, 155.3 ± 86.5	Wistar rats –suspension of CaCO_3_ in a mixture of formaldehyde and hydrogen peroxide by oral gavage administration	Gastric intraepithelial carcinomas with extensive invasion of individual tumor cells and their clustering into the muscle layer and serosa, as well as into the omentum and blood vessels (100% rats)	[146]

Abbreviations: D—diameter; L—length; Akt—protein kinase B alpha; ALP—alkaline phosphatase; ALT—alanine transaminase; AP-1—activator protein 1; AST—aspartate transaminase; BAL—bronchoalveolar lavage; BMDCs—bone-marrow-derived dendritic cells; CAC—colitis associated cancer; C/EBP—CCAAT/enhancer binding protein; CNTs—carbon nanotubes; CSCs—cancer stem-like cells; DCs—dendritic cells; DNA—deoxyribonucleic acid; Dox—doxorubicin; DSS—dextran sodium sulfate; EAE—autoimmune encephalomyelitis; EMT—epithelial-mesenchymal transition; GNFs—graphite nanofibers; nHAPs—nanohydroxyapatites; HGPRT—hypoxanthine-guanine phosphoribosyl transferase; IECs—intestinal epithelial cells; IFN—interferon; IL—interleukin; LDH—lactate dehydrogenase; LLC—Lewis lung carcinoma; MCA—methylcholanthrene; MDSCs—myeloid-derived suppressor cells; MLE—murine lung epithelial cells; MWCNR—multi-walled carbon nanotubes; NF-κB—nuclear factor kappa B; NPs—nanoparticles; OPN—osteopontin; PARP—Poly-(ADP-ribose) polymerases; PLG—Poly(lactide-co-glycolide); PD-L1—ligand 1 of programmed cell death protein; PLP—myelin proteolipid protein; PS-CaCO_3_-DNA—protamine sulfate -calcium carbonate-plasmid DNA; ROS—reactive oxygen species; SAE—small airway epithelial; SWCNT—singe-walled carbon nanotubes; TNF-α—tumor necrotic factor-α; TGF-β—transforming growth factor beta; VEGF—vascular endothelial growth factor.

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
