# Peer review of "A Brief Review about the Role of Nanomaterials, Mineral-Organic Nanoparticles, and Extra-Bone Calcification in Promoting Carcinogenesis and Tumor Progression"

_biomedicines, 2019, doi:10.3390/biomedicines7030065_

Round 1

Reviewer 1 Report

The author has written detailed review about how nanomaterials have role in carcinogenesis and tumor progression. I would like to highlight some major and minor corrections. 

Major corrections: 

A table should be provided provided listing all the essential nanomaterials (organic/inorganic) which are being discussed in the paper, their role and their limitations along with suitable references. 

Diagrams should be provided indicating how nanomaterials/ nanoparticles enter the cell, or blood vessels and how they target signaling pathways causing tumorginesis. 

A diagram or a table which shows how nanoparticles affect various body organs should be added. 

Inorganic nanoparticles have been shown to possess promising results in targeted gene/drug deliveries. Further details should be provided how they are involved in carcinogenesis. Any clinical studies relevant should be added. Apart from CA based nanoparticles, other inorganic nanoparticles should be mentioned briefly

Line 196: The PEG coated nanoparticles have also shown to enhance the drug absorption and efficiency of gene therapy. Apart from the gold NPs are there any other studies where they show distribution to other organs as well or crossing the blood brain barrier? Relevant studies should be included.

Minor corrections: 

Line 77-80 are not relevant to the prior information. 

Line 93-94: What does reduction in inflammation and fibrosis in lungs upon administration of Vitamin E diet to mice indicate? How does it counter affect the side effects of nanoparticles?

Line 240: Examples should be provided for such nanoparticles and what specific characteristics can cause induction of gaps between the endothelial cells. 

Apart from TGF beta pathway, are there any other pathways which play essential role? Are pathways such as MAPK/AKT also involved as they are essential pathways which cause tumor proliferation. Any other major upstream or downstream target molecules of essential pathways can be mentioned.

Conclusion should be put separately.

Are their any biodistribution studies performed indicating crossing of these nanomaterials to blood brain barrier? 

A small section can be added mentioning how these nanomaterials may be improved in future where the side effects can be reduced or eliminated. 

Author Response

Dear Reviewer,

Thank you for reviewing the manuscript "The role of nanomaterials, mineral-organic nanoparticles and micro-calcifications in promoting carcinogenesis and tumor progression" and valuable comments.

As for the comments:

I presented a table that summarizes the data about the main pro-carcinogenic effects of various NPs. When working on the table, I identified and corrected a number of inaccuracies in the text and references. In addition, it was discovered that a technical error was made when transferring the manuscript text to the Biomedicines Microsoft Word template file (a whole paragraph of text was missing). The text is restored, I apologize. Due to the fact that the review discusses various NPs with different compositions and sizes, different pathways and mechanisms of their entry into the body, I decided to present a picture illustrating the main pro-carcinogenic effects of NPs. Given that a very small number of studies describe in detail the changes in various organs arising under the influence of NPs, it was decided to abandon a separate figure illustrating these changes. Information about them is presented in the text of the manuscript and table 1. The problem of nanoparticles captures a huge range of issues that cannot be discussed in detail in one review. In this regard, issues related to the use of NPs in the treatment of tumors are not discussed in detail in this review. Line 77-80 removed The effect of vitamin E and other antioxidants is due to the fact that it influences one of the key mechanisms of toxic effects of NPs - to the generation of ROS. Line 240. I specified in the text which NPs and with what characteristics can cause the extravasation effect. I added to the text of the manuscript and to the references the information about the activation of MAPK and other pathway in response to the impact of LF. With this form of conclusion, I wanted to emphasize the fundamental importance of studying the role of endogenous calcium-containing nanoparticles in the induction of immunological tolerance. It is known that NPs can penetrate the blood-brain barrier, and we mention this in the manuscript. But, this issue is not discussed separately in review, since it is beyond the scope of this manuscript. Considering that a large number of various nano-objects are discussed in the review, I think that the issue of their security needs to be considered separately.

Sincerely, Dr. Marina Senchukova

Reviewer 2 Report

General

In the review article titled “The Role of Nanomaterials, Mineral-Organic Nanoparticles and Micro-Calcifications in Promoting Carcinogenesis and Tumor Progression,” author has tried to cover widely about the toxic effects of nanoparticles which can penetrate in human body through swallowing, skin penetration, or inhalation. It will be good if following suggestions are considered.

Comments:

1.       The title of the review does not match with the contents. This review covers in general pathological effects of nanoparticles and it includes brief discussion about their role in carcinogenesis. Hence, it will be better if the title is revised.

2.       There are various sentences in between which are lose and do not go with the continuity of the contents. (for example: Lines 77-80, line 277, Line 373-382). Kindly revise and provide clear understanding.

3.       The effect of various elemental nanoparticles or various organs/tissues effected should be presented in tabular form.

4.       There should be one illustration indicating harmful effects at various levels with pathological effects.

5.       The effect of nanoparticles on immune system should be discussed in separate subtopic.

Author Response

Dear Reviewer,

Thank you for reviewing the manuscript "The role of nanomaterials, mineral-organic nanoparticles and micro-calcifications in promoting carcinogenesis and tumor progression" and valuable comments.

As for the comments:

I changed the title of the manuscript to “Briefly about the Role of Nanomaterials, Mineral-Organic Nanoparticles and Extra-bone calcification in Promoting Carcinogenesis and Tumor Progression” Line 77-80 and 373-382 remuved,  I presented the table that summarizes the data about the main pro-carcinogenic effects of various NPs. When working on the table, I identified and corrected a number of inaccuracies in the text and references. In addition, it was discovered that a technical error was made when transferring the manuscript text to the Biomedicines Microsoft Word template file (a whole paragraph of text was missing). The text is restored, I apologize. In accordance with the remark, two figures were added to the manuscript. Brief information on the effect of NP to the immune system is presented in the manuscript, in Figure 1 and in the table.

Sincerely, Dr. Marina Senchukova

Round 2

Reviewer 1 Report

The author has made significant changes in the paper and added table as requested.